# Abrupt dietary changes between grass and hay alter faecal microbiota of ponies

Anna Garber [1]*, Peter Hastie[2], David McGuinness[3], Pauline Malarange[4], Jo-Anne Murray[2]

1 AB Vista, Marlborough, United Kingdom, 2 School of Veterinary Medicine, University of Glasgow, College of Medical, Veterinary and Life Sciences, Glasgow, United Kingdom, 3 University of Glasgow, Glasgow Polyomics, Glasgow, United Kingdom, 4 EPLEFPA des Combrailles, Saint Gervais D'Auvergne, Puy-de-Dôme, France

* Anna.Garber@ABVista.com

**Data Availability Statement:** All data files are available from the https://doi.org/10.5525/gla.researchdata.986

**Funding:** The authors gratefully acknowledge the scholarship (for A.Garber) and funding support for

## Abstract

Abrupt dietary changes, as can be common when managing horses, may lead to compositional changes in gut microbiota, which may result in digestive or metabolic disturbances. The aim of this study was to describe and compare the faecal microbiota of ponies abruptly changed from pasture grazing *ad libitum* to a restricted hay-only diet and *vice versa*. The experiment consisted of two, 14-day periods. Faecal samples were collected on day 0 and days 1–3,7,14 after abrupt dietary change from grass to hay and from hay to grass. Microbial populations were characterised by sequencing the V3-V4 region of the 16S rRNA gene using the Illumina MiSeq platform, 4,777,315 sequences were obtained from 6 ponies. Further analyses were performed to characterise the microbiome as well as the relative abundance of microbiota present. The results of this study suggest that the faecal microbiota of mature ponies is highly diverse, and the relative abundances of individual taxa change in response to abrupt changes in diet. The faecal microbiota of ponies maintained on a restricted amount of hay-only was similar to that of the ponies fed solely grass *ad libitum* in terms of richness and phylogenetic diversity; however, it differed significantly in terms of the relative abundances at distinct taxonomic levels. Class Bacilli, order Lactobacillales, family Lactobacillaceae, and genus *Lactobacillus* were presented in increased relative abundance on day 2 after an abrupt dietary change from hay to grass compared to all other experimental days (P <0.05). Abrupt changes from grass to hay and *vice versa* affect the faecal microbial community structure; moreover, the order of dietary change appears to have a profound effect in the first few days following the transition. An abrupt dietary change from hay to grass may represent a higher risk for gut disturbances compared to abrupt change from grass to hay.

## Introduction

The microbial population of the equine gastro-intestinal tract (GIT) is highly diverse [1–3], which may be related to the evolutionary survival/selection strategy of grazing and browsing

this study from the Trustees of the Ronald Miller Foundation. The funders had no role in study design, data collection and analysis, decision to publish, or preparation of the manuscript.

**Competing interests:** The authors have declared that no competing interests exist.

herbivores. A diverse microbial population may provide the opportunity to respond quickly to various quantities and qualities of forage available to the horse depending on seasonal and geographical factors. However, seasonal changes in feral horses happen more gradually in comparison to their domesticated counterparts. A recent study suggested that domestication of the wild horse has dramatically changed its faecal microbial composition [4]. The faecal microbiota of *Equus ferus przewalskii* was more diverse compared to domestic horses, which was attributed to the more diverse plant material that wild living horses are likely to ingest. Forage-based diets were associated with a reduction in lactic-acid bacteria and *Streptococcus bovis/equinus* and more stable microbiota in the horse [5], whilst the more diverse the microbial population is, the more adaptable it is to new feed sources. Domesticated horses are sensitive to environmental stress and may develop drastic metabolic disturbances, such as colic and laminitis in response to sudden dietary changes [6, 7]. There is clearly an association between development of colic and changes in the relative abundance of microbial taxa [8, 9]. Equine nutritionists usually recommend a gradual transition (7–14 days) [10] during any dietary change. It is, however, relatively unknown how long it takes for the gastrointestinal microbiota to adapt to a new diet. Gaining a better understanding of what bacterial populations inhabit the equine gut when horses are fed certain feedstuffs and how those populations are altered following abrupt dietary changes is important to improve gastrointestinal health of the horse.

Nowadays, there is an increasing number of studies focused on the effects of abrupt changes in the diet on hindgut and faecal microbiota in horses under different dietary conditions [11–16]. Most of those studies focus on dietary transition when rapidly fermentable carbohydrates were incorporated into the diet. Forage-only based diets appear to mainly benefit the performance of the horse. Little is known about the effect of abrupt dietary transitions of forage-only diets. Regardless of the similarity in the nutrient composition of hays, differences in hindgut microbiota were seen after the abrupt hay change, suggesting a sensitive response of the hindgut microbiota [13]. To date, there is no information available on the effects of abrupt changes from grass (pasture) to a hay-only diet, and *vice versa*. These dietary challenges mimic real-world scenarios that horse owners are likely to encounter. To the best of our knowledge, a single published article that incorporated dietary change from pasture to good-quality and poor-quality hay was predominantly focused on reporting the effect of three diets (pasture, "good hay" and "bad hay") on faecal microbiota of healthy horses and horses with asthma 3 weeks following adaptation to each diet [17]. Thus, the aim of this study was to investigate changes in equine faecal microbial population dynamics over time in response to an abrupt change from grass to hay and *vice versa*.

## Materials and methods

### Animals and experimental design

The experimental procedures in this study were approved by the University of Glasgow's School of Veterinary Medicine Veterinary Ethics & Welfare Committee (Equine hindgut health, nutrition and microbiota–Ref. 05a/14).

The effects of an abrupt dietary transition from grass to hay and *vice versa* on the faecal microbiota of ponies were investigated using Illumina next generation sequencing (NGS) methodologies. The experimental trial consisted of two, 14-day periods. Prior to commencing the study, six adult Welsh Section A gelding ponies were grazing for a one-month period (May) on the same pasture as used during the trial. The experiment began during early summer (June). Day 0 (G-D0) was recorded as the last day on pasture before the first transition to stables. Diets comprised of 100% hay (H) diet fed at 17.5 g/kg body weight (BW) on a dry matter basis. Following day 14 of the first experimental period, the second abrupt change (from

hay to grass) was implemented. During the second experimental period animals were grazed 100% on pasture (predominately ryegrass sward (*Lolium perenne*) (G) *ad libitum*. Nutritional analysis of the grass and hay were not performed. The detailed experimental design is provided in the Fig 1.

Faecal material was collected immediately (<2 minutes after being voided) between 1330 h and 1630 h on day 0 and days 1, 2, 3, 7, 14, of each experimental period using a spatula with minimum environmental contamination (spatula was sterilised between sampling from individual ponies to prevent cross contamination) and were frozen immediately (<2 minutes) at -20°C until downstream laboratory analysis.

### DNA extraction, generation of 16S amplicon libraries and sequencing

DNA extraction from faecal material was performed using Qiagen QIAamp® Fast DNA Stool Mini Kit (Qiagen, Manchester, England, UK) with slight adaptations [18]. The 16S library preparation, sequencing and data analysis were performed at Glasgow Polyomics, University of Glasgow using an adapted protocol based on the protocol provided by Illumina (San Diego, CA, USA). Nextera XT DNA Library Prep Kit was used for library preparation workflow. Briefly, 16S metagenomics protocol comprised of the main steps listed below. The V3-V4 region of the 16S rRNA gene was amplified and Illumina index primers attached in a 2-step PCR process. In the first PCR, primers targeting the V3-V4 region of the 16S rRNA gene were used: forward 5′-CTTACGGGNGGCWGCAG-3′ and reverse 5′-GACTACHVGGGTATCT AATCC-3′. The primers were designed with overhanging adapters for annealing to Illumina primers with Nextera identifier indices and sequencing adaptors. The index primers were attached during the second PCR step. Prepared libraries were stored at 4°C until use. These amplicon libraries were run simultaneously and sequenced on an Illumina MiSeq platform using an Illumina Miseq 600 cycle cartridge v3. 5% PhiX adapter-ligated library was used as a control for Illumina sequencing runs for quality control.

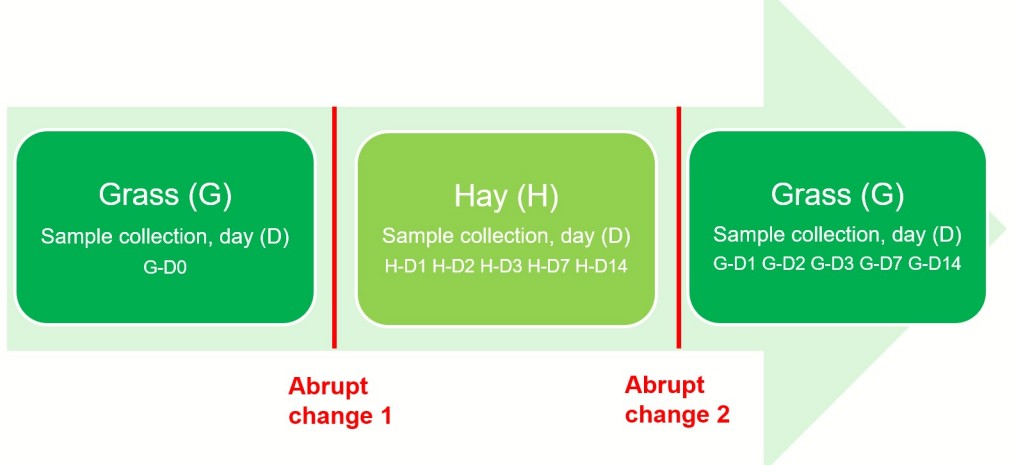

**Fig 1. Experimental design employed to study the effect of abrupt dietary changes on the faecal microbiota of ponies (n = 6).**

## Bioinformatics and statistical analysis

Each individual animal represented an experimental unit. The Illumina output was analysed with 16S metagenomics pipeline using Quantitative Insights Into Microbial Ecology (QIIME) v1.9.0 [19]. First, the *Cutadapt* Python package was used to search for the primers and adapters in all reads and remove them [20]. *Cutadapt* was also used to trim poor quality bases (using a Q-score 25 as a cut off) and to remove short reads (less than 250 bp long). Second, *Pandaseq* was used for merging paired-end reads into a single read. Third, *add_qiime_labels.py* QIIME script was used to create a single .fasta file with valid QIIME labels. Next, *pick_open_reference_otus.py* was used to perform open-reference operational taxonomic units (OTU) picking. After picking OTUs, a representative set of sequences was selected. Then, *align_seqs.py* script was used to align the sequences using PyNAST [21]. The NAST algorithm aligned each provided "candidate" sequence to the best-matching sequence in a pre-aligned database of "template" sequences. The template file was Greengenes core set (16S rRNA gene database)—available from http://greengenes.lbl.gov/. Further on, *filter_fasta.py* was used to remove chimeric sequences. Consequently, *make_otu_table.py* was used to tabulate the number of times an OTU was found in each sample and add the taxonomic predictions for each OTU. The script *make_phylogeny.py* was used to produce a phylogenetic tree relating the OTU. The script *core_diversity_analyses.py* used QIIME diversity analyses together to form a basic workflow beginning with a BIOM table, mapping file and phylogenetic tree. *Koeken.py* is a wrapper around the linear discriminant analysis effect size (LEfSe) script, it was used to create a convenient workflow from a QIIME OTU table to biomarker microbial taxa.

LEfSe is a tool [22] which was used to find biomarkers specific to each collection day using relative abundance (RA). Firstly, the non-parametric factorial Kruskal-Wallis sum-rank test was used to detect features with significant differential abundance with respect to the class of interest; subsequently, a set of pairwise tests among subclasses was carried out using the unpaired Wilcoxon rank-sum test. As a last step, LEfSe used linear discriminant analysis to estimate the effect size of each differentially abundant feature. The differences in microbial RA were considered significant when LDA score (log10) >2 [22]. LEfSe was performed using the OTU table with all the taxonomic levels and day of sampling as class.

α –diversity measures were used to show which microbial populations were the most compositionally diverse and which had most phylotypes or branch lengths [23, 24]. OTU or phylotype, is usually used instead of "species" for a cluster of related 16S rRNA sequences in 16S rRNA sequencing studies [25].

α –diversity of the microbial communities was evaluated using Chao1 richness non-parametric estimator (non-phylogenetic metric), PD_whole_tree (phylogenetic metric) and observed OTUs. The PD_whole_tree metric, also known as Faith's Phylogenetic Diversity, is based on the phylogenetic tree. It sums up all the branch lengths as a measure of diversity: if a new OTU is closely related to another OTU in the sample, there will be a small increase in diversity. Conversely, if a new OTU originates from a completely different lineage compared to all other OTUs in the sample, it will contribute significantly to increase in the diversity.

This 16S rRNA gene sequencing technique does not use reads long enough to identify bacterial and archaeal sequences down to species taxonomic level. Hence, any OTUs which were revealed to be significantly different in relation to diet were traced back to their corresponding genera.

To analyse β-diversity, weighted and unweighted UniFrac metrics in QIIME were used [26, 27]. β-diversity metric calculates the distance between a pair of samples and shows which factors correlate with differences in microbiota composition. In weighted UniFrac branch length were weighed by relative abundance, while unweighted UniFrac were based purely on

sequence distance measure. UniFrac distance was calculated by constructing a phylogenetic tree from all the OTUs [27]. For each pair of samples, the distance was estimated as a (sum of unshared branch lengths)/ (sum of total branch lengths).

*principal_coordinates.py* and *make_emperor.py* were used to create principal coordinate analysis (PCoA) plots and to generate charts [28]. Clustering of samples was evaluated by plotting the resultant vector of the PCoA with 3 dimensions. The false discovery rate (FDR) P-value was used to adjust for multiple testing (Benjamini-Hochberg procedure). FDR P-value cutoff was preset at <0.1.

## Results

### Sequencing quality

Faecal samples (n = 66) from 6 ponies on 11 collection days were sequenced and generated 4,895,038 total reads. A total of 4,777,315 reads for the pooled microbial communities (bacterial and archaeal), which passed through quality filters, were approved for further analysis. The reads were assigned into OTU and 2,232 OTU observations in total were obtained with the counts of reads per sample (mean ± SD) 72,384 ± 14,322. A rarefaction depth of 2,000 reads per sample was used in order to normalise the number of reads across all samples. The sub-sampling size was considered adequate, as evidenced by the plateau of rarefaction curves (S1–S3 Figs). OTUs with less than 100 reads were excluded from the analysis, leaving 2,221 OTUs in total.

### α-diversity

There were no significant differences in phylogenetic diversity (PD_whole_tree), estimated total species richness (Chao_1), and richness expressed as the number of observed OTUs (Observed_OTUs) between different sampling days (S1–S3 Figs). However, it can be seen from Fig 2, that for H-D1 and G-D14 Chao1 diversity metrics were substantially higher and

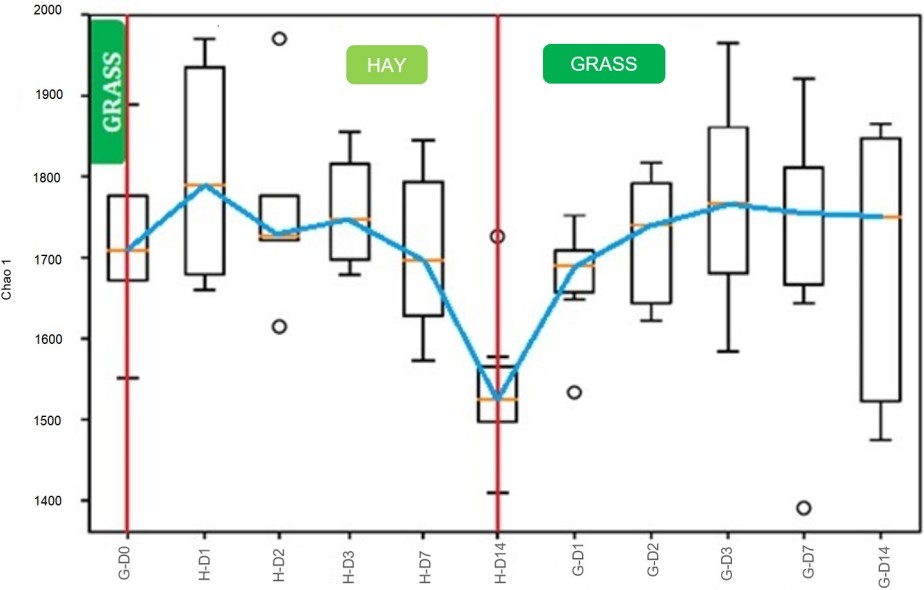

**Fig 2. Chao1 for bacterial communities in the faeces of the ponies for different days of experiment.** Boxplot comparing estimated richness (n = 6). Red bars show abrupt changes: 1st from grass (G) to hay (H), 2nd from hay to grass. Blue line connects the means (orange lines). Circles represent outliers. D: days.

showed greater variance compared to H-D14 (H-D1 95% CI = 898, 1181; H-D14 95% CI = 737, 1033).

A PD_whole tree (S4 Fig) revealed that the microbial diversity was different between the following pairs of animals: pony 1 and pony 5, pony 2 and pony 4, pony 4 and pony 5, and pony 5 and pony 6 (P = 0.015).

### β-diversity

Principal coordinate analysis on a phylogenetic weighted UniFrac dissimilarity matrix revealed no clustering of equine faecal samples by dietary group when all experimental days were compared (Fig 3A); however, there was clustering observed on H-D1 and G-D1 (Fig 3B) and H-D3 and G-D3 (Fig 3C) with more than half (52.19%) of the variation explained by principal coordinates (PC) 1, 2 and 3. Meanwhile, there was no clustering of equine faecal samples by dietary group observed on day 7 or 14 after the abrupt dietary change.

Principal coordinate analysis on an unweighted-UniFrac dissimilarity matrix revealed no clustering of equine faecal samples by dietary group on any of the sampling days, with only 18.40% of the variation explained by principal coordinates (PC) 1, 2 and 3 (Fig 4).

### Relative abundance at various taxonomic levels

Within the domain Bacteria there were 18 phyla identified, while there was only 1 phylum detected within the domain Archaea. Bacterial phyla encompassed 32 different classes, 41 orders, 65 families and 109 genera. Archaeal phylum incorporated 3 different classes, 4 orders, 4 families and 5 genera. The detailed structure of the microbial populations is provided in S1 Table. Several microorganisms were detected, which have not been previously assigned in the Greengenes database.

LEfSe analysis revealed 46 biomarkers, associated with sample collection days (G-D0, H-D1, H-D2, H-D3, H-D7, H-D14, G-D1, G-D2, H-D3, H-D7, H-D14) (LDA score >2, P <0.05). Overall, there were 1 phylum, 4 classes, 6 orders, 11 families and 24 genera identified as biomarkers for one of the sample collection days and those are listed in Tables 1 and 2.

Though there was no clustering of equine faecal samples by dietary group on day 14 observed with PCoA, LEfSe analysis revealed 63 biomarkers that were different on day 14

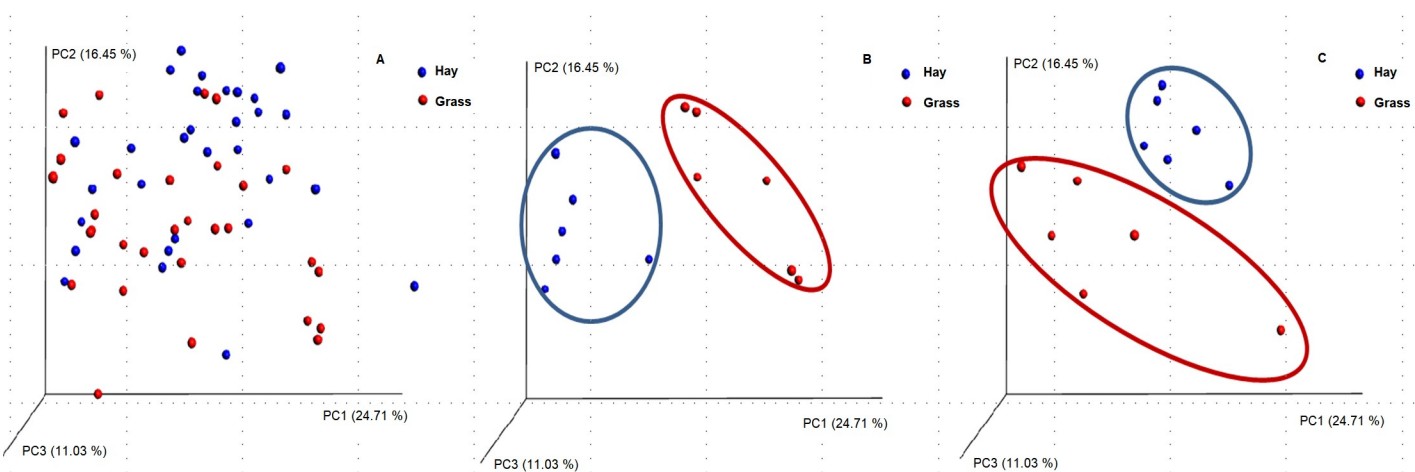

**Fig 3. PCA on a weighted-UniFrac dissimilarity matrix for data on the microbial membership and structure in the faeces of ponies.** (A) Days 1, 2, 3, 7, 14 on grass and hay (H-D1, H-D2, H-D3, H-D7, H-D14, G-D1, G-D2, G-D3, G-D7, G-D14) (B) Day 1 after abrupt change on grass and hay (H-D1, G-D1) (C) Day 3 after abrupt change on grass and hay (H-D3, G-D3). OTUs with <100 reads were excluded from the analysis. n = 6. H: hay; G: grass; D: day; PCA: Principal coordinate analysis.

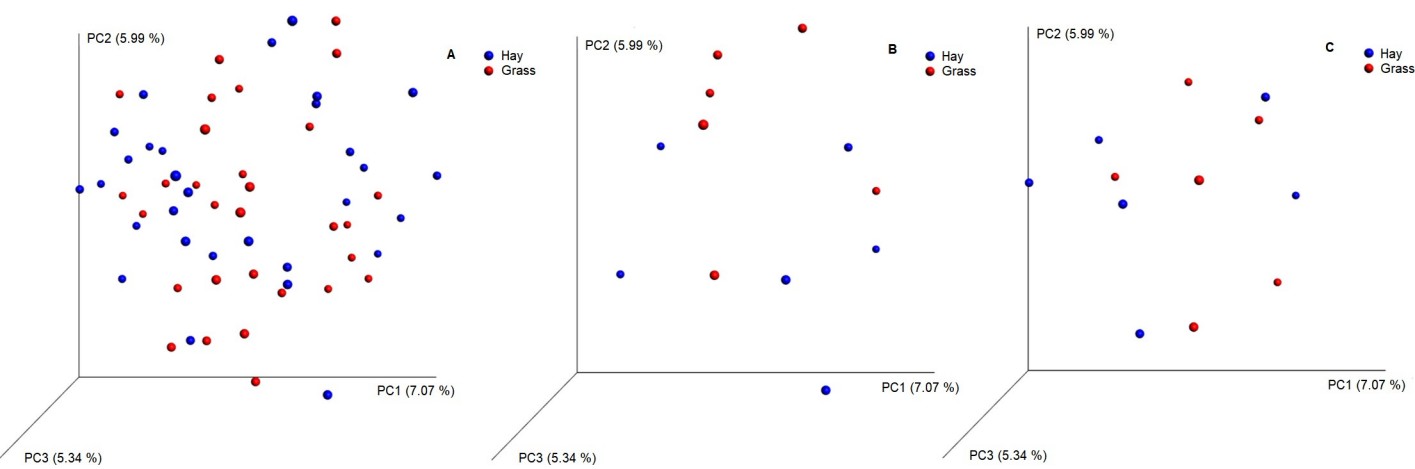

**Fig 4. PCA on an unweighted-UniFrac dissimilarity matrix for data on the microbial membership and structure in the faeces of ponies.** (A) Days 1, 2, 3, 7, 14 on grass and hay (H-D1, H-D2, H-D3, H-D7, H-D14, G-D1, G-D2, G-D3, G-D7, G-D14) (B) Day 1 after abrupt change on grass and hay (H-D1, G-D1) (C) Day 3 after abrupt change on grass and hay (H-D3, G-D3). OTUs with <100 reads were excluded from the analysis. n = 6. H: hay; G: grass; D: day; PCA: Principal coordinate analysis.

between grass and hay diets (LDA score >2, P <0.05). Overall, there were 2 phyla, 3 classes, 9 orders, 18 families and 31 genera that were identified as different on day 14 when ponies were receiving hay or grass diets (H-D14 and G-D14).

**Table 1. Linear discriminate analysis effect size (LEfSe) showing significant taxa (biomarkers, in bold) in the faeces of the ponies for each sampling day (D) when ponies (n = 6) were maintained on hay (H).**

| Phylum>Class>Order>Family>Genus | LDA | P-value | Biomarker |
|---|---|---|---|
| Firmicutes>Clostridia>Clostridiales>Lachnospiraceae>**Shuttleworthia** | 4.9 | 0.004 | D1-H |
| Firmicutes>Clostridia>Clostridiales>Lachnospiraceae>**Butyrivibrio** | 4.1 | 0.005 | D2-H |
| Verrucomicrobia>**Opitutae** | 4.4 | <0.001 | D2-H |
| >_**Cerasicoccales**_ | 4.5 | <0.001 | D2-H |
| >_**Cerasicoccaceae**_ | 4.6 | <0.001 | D2-H |
| >* | 4.3 | <0.001 | D2-H |
| Firmicutes>Clostridia>Clostridiales>Syntrophomonadaceae>**Syntrophomonas** | 4.5 | 0.044 | D3-H |
| >**Veillonellaceae** | 5.9 | 0.003 | D3-H |
| >* | 5.3 | 0.002 | D3-H |
| >**Phascolarctobacterium** | 5.8 | 0.003 | D3-H |
| >**Syntrophomonadaceae** | 4.5 | 0.044 | D3-H |
| >Clostridiaceae>* | 4.9 | 0.049 | D3-H |
| >**Lachnospiraceae** | 6.6 | 0.029 | D3-H |
| >Lachnospiraceae>* | 6.5 | 0.038 | D3-H |
| >Lachnospiraceae>**Dorea** | 4.5 | 0.011 | D3-H |
| >Lachnospiraceae>**Blautia** | 5.7 | 0.005 | D3-H |
| >Lachnospiraceae>**Anaerostipes** | 5.0 | 0.024 | D3-H |
| Firmicutes>Erysipelotrichi>Erysipelotrichales>Erysipelotrichaceae>* | 5.0 | <0.001 | D14-H |
| > _**Other** | 4.5 | 0.001 | D14-H |
| Firmicutes>Clostridia>Clostridiales>**Gracilibacteraceae** | 4.4 | 0.043 | D14-H |
| >**Lutispora** | 4.4 | 0.043 | D14-H |

* indicates unclassified taxonomic level. LDA: linear discriminate analysis score.

**Table 2. Linear discriminate analysis effect size (LEfSe) showing significant taxa (biomarkers, in bold) in the faeces of the ponies for each sampling day (D) when ponies (n = 6) were maintained on grass (G).**

| Phylum>Class>Order>Family>Genus | LDA | P-value | Biomarker |
|---|---|---|---|
| Bacteroidetes>Bacteroidia>Bacteroidales>* | 7.1 | 0.010 | D1-G |
| Bacteroidetes>Bacteroidia>Bacteroidales>*>* | 7.1 | 0.010 | D1-G |
| >_Paraprevotellaceae_>**CF231** | 5.8 | 0.038 | D1-G |
| Proteobacteria>Deltaproteobacteria>**Desulfovibrionales** | 4.8 | 0.039 | D1-G |
| >**Desulfovibrionaceae** | 4.8 | 0.039 | D1-G |
| Firmicutes>Clostridia>Clostridiales>Lachnospiraceae>**Roseburia** | 4.8 | 0.004 | D1-G |
| **Spirochaetes** | 6.7 | 0.002 | D1-G |
| >**Spirochaetes** | 6.7 | 0.001 | D1-G |
| >**Spirochaetales** | 6.7 | 0.001 | D1-G |
| >**Spirochaetaceae** | 6.7 | 0.001 | D1-G |
| >**Treponema** | 6.7 | 0.001 | D1-G |
| Cyanobacteria>**Chloroplast** | 4.8 | 0.014 | D2-G |
| >**Streptophyta** | 5.0 | 0.014 | D2-G |
| >* | 5.0 | 0.014 | D2-G |
| >* | 5.0 | 0.014 | D2-G |
| Firmicutes>Clostridia>Clostridiales>Clostridiaceae>**Sarcina** | 4.4 | <0.001 | D2-G |
| >Lachnospiraceae>**Lachnobacterium** | 4.6 | 0.017 | D2-G |
| Firmicutes>**Bacilli** | 4.5 | 0.016 | D2-G |
| >**Lactobacillales** | 4.5 | 0.016 | D2-G |
| >**Lactobacillaceae** | 4.5 | 0.016 | D2-G |
| >**Lactobacillus** | 4.5 | 0.016 | D2-G |
| Actinobacteria>Coriobacteriia>Coriobacteriales>Coriobacteriaceae>**Slackia** | 4.5 | 0.002 | D2-G |
| Proteobacteria>Alphaproteobacteria>Rickettsiales>**mitochondria_Other** | 4.7 | 0.006 | D3-G |
| Firmicutes>Clostridia>**Clostridiales_Other_Other** | 5.6 | 0.017 | D3-G |
| Actinobacteria>Coriobacteriia>Coriobacteriales>Coriobacteriaceae>**Adlercreutzia** | 5.3 | 0.020 | D3-G |

* indicates unclassified taxonomic level. D: day; H: hay; G: grass; LDA: linear discriminate analysis score.

**Phylum level.** There were 19 phyla identified among 66 samples. Within the domain Bacteria there were 7 phyla identified in the faeces of the ponies, with an average relative abundance across all samples of >1%. LEfSe identified significant differences in phylum Actinobacteria and phylum Spirochaetes on H-D14 and G-D14 (LDA score >2, P <0.05) (S5 Fig).

Two phyla, Bacteroidetes and Firmicutes dominated the bacterial community regardless of the diet or sampling time. Interestingly, on G-D0 (after ponies were grazing for a month before the first experimental period began), H-D14 and G-D14, Bacteroidetes was the most abundant phyla with Firmicutes being the second. However, on H-D1, H-D2, H-D3, G-D2 and G-D3 Firmicutes was identified as the most abundant phylum followed by Bacteroidetes (Fig 5).

**Genus level.** There were 114 genera identified among 66 samples. There were 7 genera with an average RA across all samples higher than 5%; these 7 genera represented 73% of the samples' genus RA on average. There were 26 genera identified with a RA >0.5% and, with the exception of unassigned genus from family Paraprevotellaceae, these 26 most abundant genera were present in the samples from all ponies regardless of diet or sampling day and represented 94% of the total RA at genus level. Among those genera were *Fibrobacter, Ruminococcus, Treponema, Paludibacter, Prevotella, YRC22, CF231, BF311, Clostridium, RFN20, Blautia, and Phascolarctobacterium.*

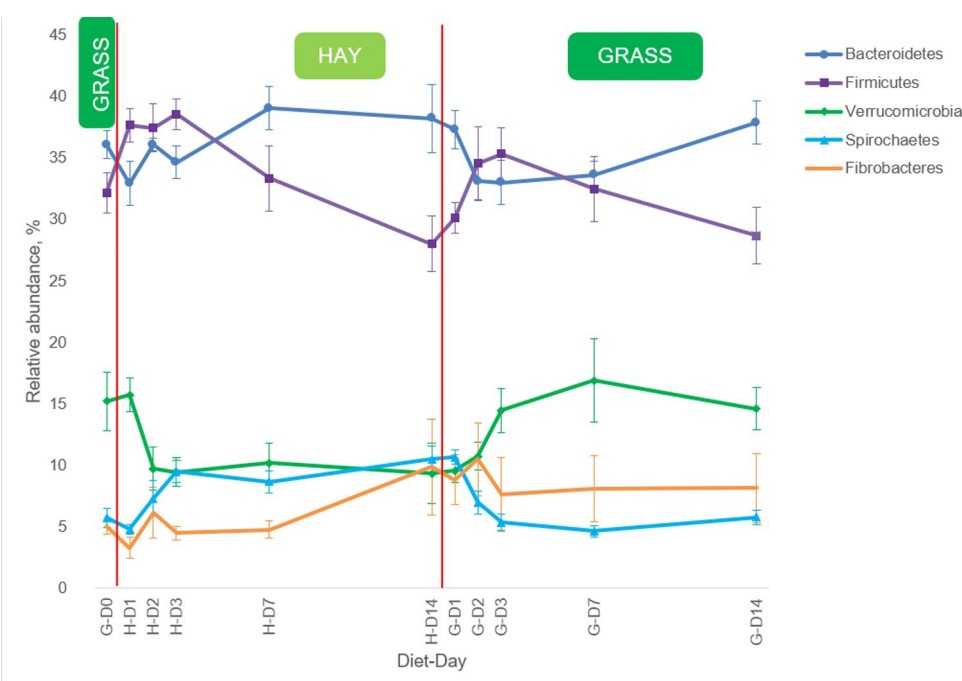

**Fig 5. Bacterial phyla dynamics in the faeces of the ponies during the abrupt dietary transitions.** Bacterial phyla with relative abundance >5% are presented. Error bars represent standard error of the mean. Red vertical lines represent abrupt changes in the diet. D: day; H: hay; G: grass.

The most abundant genus belonged to the phylum Bacteroidetes, class Bacteroidia, order Bacteroidales unassigned at family and genus level and its RA across all samples was 22.8 ± 0.62% (mean ± SEM). The dynamics of relative abundance in the 1st to 7th most abundant genera (average RA across all sampling days >5%) after abrupt changes in the diets is depicted in Fig 6.

## Discussion

This study was conducted to evaluate whether an abrupt change from grass to hay and *vice versa* would have an impact on faecal microbial communities in ponies. Such changes have been previously reported to be a risk factor for digestive upsets and colic manifestations in epidemiological studies [29, 30].

Alpha-diversity metrics were not different between diets and experimental days. This indicated that the differences observed in the PCoA plots were likely due to differences in the RA of some taxonomic levels, as the numbers of taxa and the evenness did not change significantly between faecal samples from ponies fed either diet. This does not, however, exclude the possibility of a radical shift in taxa without an overall change in taxa numbers or evenness, although the taxonomic distribution strongly suggests this is not the case. This is in agreement with the results reported by other researchers [12], and may be due to both types of forage (grass and hay) generally having a substantial fibre content, which is beneficial for the proliferation of fibrolytic bacteria: crude fibre—330 g/kg and 264 g/kg, NDF– 670 g/kg and 400–570 g/kg for grass hay and grass, respectively [31].

The clustering observed in the first few days after the dietary change suggests that a shorter more intense sampling period around the time of dietary transition should be considered. Future studies on the faecal microbiota of horses should consider more frequent sampling to

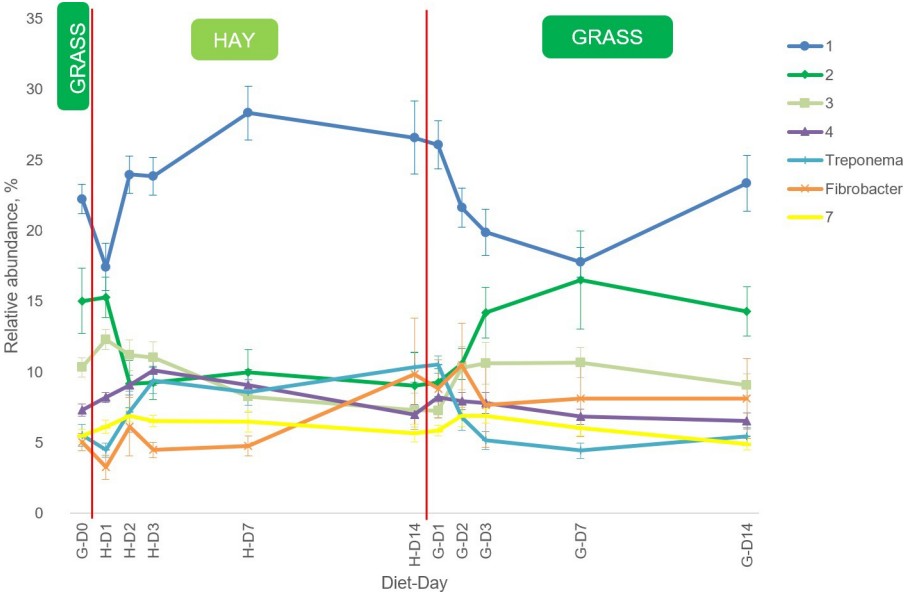

**Fig 6. Bacterial genera dynamics in the faeces of the ponies (n = 6) during the abrupt transitions.** Bacterial genera with average relative abundance (RA) >5% are presented. Error bars represent standard error of the mean. Red vertical lines represent abrupt changes in the diet. Unassigned genera were numbered in descending order based on their RA. D: day; H: hay; G: grass. All genera were traced back to their correspondent phylum > class > order > family > genus. 1: Bacteroidetes > Bacteroidia > Bacteroidales > unassigned > unassigned. 2: Verrucomicrobia > Verruco-5 > WCHB1-41 > RFP12 > unassigned. 3: Firmicutes > Clostridia > Clostridiales > Ruminococcaceae > unassigned. 4: Firmicutes > Clostridia > Clostridiales > Lachnospiraceae > unassigned. Treponema: Spirochaetes > Spirochaetes > Spirochaetales > Spirochaetaceae > Treponema. Fibrobacter: Fibrobacteres > Fibrobacteria > Fibrobacterales > Fibrobacteraceae > Fibrobacter. 7: Firmicutes > Clostridia > Clostridiales > unassigned > unassigned.

gain better understanding of when any structural adaptation takes place. Furthermore, it is important to consider the differences in compositional adaptation and functional adaptation of gut microbiota. Compositional adaptation (alterations in structure) is known to be rapid, while functional adaptation (full adaptation to the new diet) may take place over an extended period of time. Previous research suggests that horse gut microbiota can adapt in response to new diets quickly within 4–6 days [12, 32]. This is in agreement with the results of the current study where according to PCoA and LEfSe, most of the differences in microbial community structure were observed during the first 3 days after dietary change. These results suggest a rapid microbial community response. Rapid adaptation to dietary changes have also been reported in human research. David, Maurice *et al*., reported that the human gut microbiome can revert to its original structure within two to four days post dietary transition from a plant-based diet to an animal-based diet or *vice versa*, which may reflect selection processes of human evolution [33]. However, there are an insufficient number of studies reporting functional adaptation of the equine gut microbiome to new dietary and environmental conditions. The change between two silages with different crude protein content resulted in altered colonic bacterial counts within the first 24 h after the abrupt change was introduced [15]. Moreover, days 7 to 22 after abrupt change were associated with a decrease in pH and an increase in volatile fatty acids concentration [15]. Hence, both short-term and long-term changes may occur in response to abrupt dietary changes in equine diets. Long-term functional changes require further investigation.

A recent study by Leclere and Costa [17] examined the effect of pasture, high-quality and poor-quality hay on healthy horses and horses with asthma. There were 2 genera

overrepresented in faeces of the healthy horses maintained on pasture (*Coprococcus* and unclassified genus from family Enterobacteriaceae), 4 on good-quality hay (*Denitrobacterium*, *Cellulosilyticum*, unclassified genus belonging to Deltaproteobacteria class and unclassified genus belonging to order Sphingobacteriales), and none on dusty hay. However, the effect of abrupt dietary change from pasture to hay or *vice versa* was not discussed in the aforementioned study as samples were collected 3 weeks after diets were introduced to horses.

The current study found Bacteroidetes was the most abundant phylum in the faeces of ponies maintained on hay or grass on H-D7, H-D14, G-D7, and G-D14, closely followed by Firmicutes. However, short term after abrupt dietary change Firmicutes phylum generally dominated. This finding may suggest that Firmicutes are more adept at adjusting to abrupt dietary changes regardless of the order, compared to Bacteroidetes. Furthermore, it can be speculated that the Bacteroidetes:Firmicutes ratio was altered not by the diet itself (hay or grass), but by the abrupt change introduced to the diet. The proportions of Firmicutes and Bacteroidetes have been shown to be indicators of digestive health in humans [34], where obese people had fewer Bacteroidetes and more Firmicutes than their leaner counterparts. The same pattern was observed when obese and lean horses were studied [35]. Costa, Arroyo [36] reported that in healthy horses Firmicutes predominated (68%) followed by Bacteroidetes (14%), while in horses with colitis Bacteroidetes (40%) was the most abundant phylum followed by Firmicutes (30%). It appears that the Firmicutes to Bacteroidetes ratio might be an important predictor of metabolic changes in the GIT of the horse. The existing knowledge is, however, limited and does not allow us to interpret an increase or decrease in Firmicutes to Bacteroidetes ratio as a positive or negative change. The results of the current study are in disagreement with the findings of Grimm, Philippeau [14] who observed an increase in Bacteroidetes in the caecum of the horse on day 1 after abrupt dietary changes of two different hays. This may be due to different diets used in both studies, using faeces as a source of microbial exploration as opposed to caecum contents, as well as possible inter-breed variations, indicating that further research is required to validate current findings.

In the current study, the RA of faecal class Bacilli, order Lactobacillales, family Lactobacillaceae and genus *Lactobacillus* was increased on day 2 after the abrupt dietary change from hay to grass compared to all other sampling days. This finding suggests that an abrupt dietary change from hay to grass may represent a higher risk for hindgut pH to drop compared to abrupt change from grass to hay as *Lactobacillus* is known to be a major lactic acid producing bacteria. Conversely, in the study of Muhonen, Julliand [16] a change from hay to haylage or silage from the same swath harvested on the same date did not induce alterations in bacterial counts after 28 hours. However, volatile fatty acids and pH were associated with a modification of the colonic *lactobacilli* and *streptococci* bacterial counts on days 8 and 21 after the abrupt change took place, suggesting a long-term alteration in the hindgut environment, rather than the short-term changes observed in the current study.

There were differences observed between samples from each individual pony. Significant inter-horse variation in gut microbiota has been previously reported [37–39]. Regardless of the changes observed in the faecal microbiota of ponies as a result of the abrupt change in diet, none of the animals in this current study developed any clinical signs of digestive upset. This finding suggests that compositional differences in gut microbiota alone cannot explain the development of metabolic disorders, such as colic and laminitis, but may provide insights into potential mechanisms of development of these conditions. This hypothesis warrants further investigation using a larger animal sample size. It is likely that the adaptive capacity of equine gut microbiota in response to dietary changes depends on each individual animal. This might explain why some horses are more prone to the development of metabolic disorders in response to similar dietary changes compared to other individuals. More research on

individual traits that may predispose horses to the development of metabolic disorders would be an essential next step for further work.

The ponies used in the current study were of the same sex and breed and were maintained on the same farm and fed the same diets during the experimental period. This is in contrast to other microbiome studies that involved horses maintained on various diets and managed differently [1, 39–41]. It is possible that by reducing individual animal variation under controlled conditions in this relatively small sample size, has enabled us to identify the differences truly related to the abrupt change from grass to hay and *vice versa*.

In the current experiment grass was fed *ad libitum* while hay intake was restricted to 17.5 g/kg BW on dry matter basis. Hence, this study investigated not only the change of the diets but also incorporated a change in the amount of feed offered. This choice was made with the purpose of replicating the real-world scenario by which horse owners are likely to manage their horses. Moreover, it is important to remember that the change was not only related to diet, but a complex interaction of social-behaviour change, change in exercise, climate effects attributed to differences in management of the stabled horse and horse maintained on pasture. Furthermore, some of the bacterial community structural changes observed in the grazing ponies may be a result of changes in the composition of grass and environmental conditions. Weather conditions, such as temperature and rainfall, influence horse faecal microbiota [42]. This effect can be attributed to the direct influence of weather on horses or an indirect influence of weather or soil microbiota and other environmental bacteria, which after being ingested, may affect equine gut microbiota. A limitation of this study design is the small sample size (6 animals) and lack of control group that remained on pasture. As mentioned previously, variables such as temperature, pasture composition, change in environment (housing) could have had an effect on their microbiota. If environmental factors were majorly responsible for the observed changes in microbial relative abundance, the microbiota of G-D14 would be most likely to be dissimilar to G-D0. In our study, according to LEfSe analysis there were no biomarkers identified for G-D0 or G-D14. According to weighted and unweighted UniFrac PCoA plots, there was no clustering observed on G-D0 and G-D14 (S6 Fig). Thus, it is likely that abrupt dietary changes were major factors leading to changes in faecal microbiota of the ponies abruptly transitioned from grass to hay and from hay to grass.

The OTU is a vague term that may be defined as an individual organism, a named taxonomic group such as a species or genus, or a group with undetermined evolutionary relationships that share a given set of observed characters [43]. In NGS studies, an OTU is typically defined as a cluster of reads with 97% similarity, expecting that these correspond approximately to species [44]. Correspondence of OTUs with species may fail because of various factors. Firstly, some species have genes that are >97% similar, resulting in merged OTU containing multiple species. Secondly, a single species may have paralogs that are <97% similar, causing the species to be split across two or more OTUs. Finally, some clusters may be false due to artefacts including read errors and chimeras. Due to the concerns discussed above, an exact correspondence between OTU and species cannot be determined. Therefore, when the Greengenes database attempted to place reads into a phylogenetic tree and then assign names to parts of that tree, sequences belonging to mitochondria and chloroplast rRNA were dispositioned. For the purposes of this study, the full taxonomy of the mitochondria and chloroplast reads were ignored, as they were from mitochondria and chloroplast and did not evolve from the Bacteria kingdom.

Abrupt changes from grass to hay and *vice versa* affected the faecal microbial community structure. Moreover, the order of dietary change had a profound effect in the first few days following the transition. There is a requirement to evaluate nutritional strategies with the potential to mitigate the effect of abrupt dietary changes on GIT microbiota. Clearly, further studies

are required to identify a tool for supporting a more stable GIT environment in the horse during abrupt dietary transitions. Determining the role of microbial species in nutrient breakdown and their contribution to metabolism is fundamental to understanding digestive processes and maximising productivity, health and welfare of the horse; however, 16S rRNA gene sequencing only allows us to identify compositional changes in microbial populations. Results from this study can be used to further investigate the role of specific microbes that were affected by the abrupt dietary changes and their contribution to equine microbiota in terms of their immediate response to the change and longer-term adaptation to different diets.

## Supporting information

**S1 Fig. Rarefaction curves for bacterial communities in the faeces of the ponies (n = 6) showing α-diversity metric PD_whole_tree as a function of sequences per sample.** Each curve represents sampling day (D) on grass (G) or hay (H).
(TIF)

**S2 Fig. Rarefaction curves for bacterial communities in the faeces of the ponies (n = 6) showing α-diversity metric Chao1 as a function of sequences per sample.** Each curve represents sampling day (D) on grass (G) or hay (H).
(TIF)

**S3 Fig. Rarefaction curves for bacterial communities in the faeces of the ponies (n = 6) showing α-diversity metric observed OTUs as a function of sequences per sample.** Each curve represents sampling day (D) on grass (G) or hay (H).
(TIF)

**S4 Fig. Boxplots comparing faecal microbial α-diversity metric (PD_whole_tree) for each individual pony (P).** n: number of samples per pony.
(TIF)

**S5 Fig. Differential features in the faeces of the ponies (n = 6) fed grass and hay diets on day 14 (H-D14 and G-D14) at phylum level.** Solid and dashed horizontal lines indicate mean and median across all samples, respectively.
(TIF)

**S6 Fig.** 3-dimensional Principal Coordinate Analyses plot (on a phylogenetic weighted (A) unweighted (B) UniFrac dissimilarity matrix) for data on the microbial membership and structure in the faeces of ponies (n = 6) across H-D14 and G-D14. There was no clustering observed. OTUs with <100 reads were excluded from the analysis. H: hay; G: grass; D: day.
(TIF)

**S1 Table. Faecal bacterial and archaeal communities of mature Welsh Section A geldings (n = 6) during the study investigating abrupt change of the hay and grass diets.** The bacterial and archaeal taxa identified in the faeces of ponies in the present study are listed in the table according to the taxonomic ranks using Greengenes database.
(DOCX)

## Author Contributions

**Conceptualization:** Anna Garber, Peter Hastie, Jo-Anne Murray.

**Data curation:** Anna Garber, Pauline Malarange.

**Formal analysis:** Anna Garber, David McGuinness.

**Funding acquisition:** Peter Hastie, Jo-Anne Murray.

**Investigation:** Anna Garber, Pauline Malarange.

**Methodology:** Anna Garber, David McGuinness, Jo-Anne Murray.

**Project administration:** Anna Garber, Peter Hastie, Jo-Anne Murray.

**Resources:** Peter Hastie, Jo-Anne Murray.

**Software:** David McGuinness.

**Supervision:** Peter Hastie, Jo-Anne Murray.

**Validation:** Peter Hastie, Jo-Anne Murray.

**Visualization:** Anna Garber, David McGuinness.

**Writing – original draft:** Anna Garber.

**Writing – review & editing:** Peter Hastie, Jo-Anne Murray.

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
