## [Decision Letter · Decision Letter 0]

27 May 2020

PONE-D-20-08059

Abrupt dietary changes between grass and hay alter faecal microbiota of ponies

PLOS ONE

Dear Dr. Garber,

Thank you for submitting your manuscript to PLOS ONE. After careful consideration, we feel that it has merit but does not fully meet PLOS ONE’s publication criteria as it currently stands. Therefore, we invite you to submit a revised version of the manuscript that addresses the points raised during the review process.

We look forward to receiving your revised manuscript.

Kind regards,

Juan J Loor

Academic Editor

PLOS ONE

Journal Requirements:

2. We noted in your submission details that a portion of your manuscript may have been presented or published elsewhere:

'An abstract based on the part of the current study was presented at EWEN (European Workshop in Equine Nutrition) 2018 in Sweden as oral presentation and published in the proceedings. This research study is a part of PhD thesis which is currently embargoed for 2 years. '

Please clarify whether this conference proceeding was peer-reviewed and formally published.

If this work was previously peer-reviewed and published, in the cover letter please provide the reason that this work does not constitute dual publication and should be included in the current manuscript.

3. We note you have included tables to which you do not refer in the text of your manuscript. Please ensure that you refer to Tables 3 and 4 in your text; if accepted, production will need this reference to link the reader to the Table.

Reviewers' comments:

Reviewer's Responses to Questions

**Comments to the Author**

1. Is the manuscript technically sound, and do the data support the conclusions?

Reviewer #1: Yes

2. Has the statistical analysis been performed appropriately and rigorously? 

Reviewer #1: Yes

3. Have the authors made all data underlying the findings in their manuscript fully available?

Reviewer #1: No

4. Is the manuscript presented in an intelligible fashion and written in standard English?

Reviewer #1: No

5. Review Comments to the Author

Reviewer #1: Dear Editor and authors.

This is a descriptive study evaluating changes in the fecal microbiota of 6 ponies undergoing diet change. The article is not completely novel, but brings interesting information to contribute to the current knowledge of factors affecting the intestinal microbiota of horses. The major limitation with the study design is the lack of control group that remained in pasture: variables such as temperature, pasture composition, change in environment (housing) could have affected their microbiota. This needs to be clearely addressed in the discussion. Also, authors give too much importance for the microbial variations at the phylum level, which is important, much less informative and meaningful. For example, which of the thousands species of Firmicutes increased after diet change? These genera should be presented in the main article (and not as supplementary data). As you will see in my comments below the article needs major rewriting to improve clarity.

Abstract:

The abstract needs major changes. You need to clearly state the study design including days of sampling, details on diet change, number of animals, etc. Many readers are not familiarized with concepts like LEfSe, LDA, PCoA, community structure, etc. You should use this space wisely to transmit the main findings and significance of your study in a way that most people will be able to understand it.

Line 53-56: I am not sure this statement should be used to justify your study. While fiber based diets likely benefit the horse microbiota, this is not feasible in performance horses in which high energy supplementation is required to fulfill their energy needs.

Line 59: please refer to Leclere and Costa, J Vet Intern Med. 2020 Mar;34(2):996-1006. doi: 10.1111/jvim.15748.

Material and methods:

What was the composition and quality of hay (any analysis done?)? What was the grass type/variety?

Table 1: it would be better to see that in a timeline rather than in a Table. You should not use the same nomenclature (e.g. D1) for both periods (e.g. line 170, 209, 294, etc.). In fact, every time you write “abrupt dietary change” you should specify to which of the changes you are referring to.

Line 130: what is a phylotype?

Line 132: what is a OTU? Please define those terms. Did you use a phylotype or OTU approach?

Line 147: adjusted using which test?

Can you comment on the reliability of those primers on classifying Archaea based on a scientific publication? It is my impression that primers designed to target bacteria are not adequate for Archaea analysis.

Line 157: why were low abundance sequences excluded? Was this done before or after alfa diversity calculation? This will likely impact Chao index.

Results:

Line 160: is phylogenetic diversity a measure of Alpha diversity? Please indicate which test was used for that in the material and methods.

Line 163: Please indicate the units after the values.

Please provide details how you performed LEfSe: using the OTU table with all the taxonomic levels? using day of sampling as class (or samples from G and H analysed separately, or performed pairwise comparisons?). Wouldn’t make more sense to use only the different genera?

Results:

Line 154: counts of reads per sample or of OTUs per sample?

Line 157: where are the rarefaction curves? This analysis can be quite subjective and I recommend that Good’s coverage is presented instead.

Figure 1: could indicate the diets and day of change, just like you did in Figure 3. Please indicate what circles represent (outliers?).

Line 177: is the weighted Unifrac really evaluating microbial membership and structure or only structure. In my understanding, membership is evaluated by unweighted Unifrac. In fact, it would also be interesting to see results from weighted unifrac that are as important as the ones presented here.

Line 180: OTUs

Figure 2: how can you distinguish the different days in figure 2A? Any clustering seen by individual?

Line 136: you mentioned that Illumina sequencing is not suitable for species level analysis (and you are completely right), so in my opinion it is not adequate to present this data, as it might mislead readers to erroneous information.

Line 188: were those really unassigned organisms or it could be products of sequencing error not detected during bioinformatics? This should be acknowledged especially considering the approach adopted here (sequencing of the V3-V4 regions), which considerably decreases the overlapping and therefore, increases sequencing errors:

https://mothur.org/blog/2014/Why-such-a-large-distance-matrix/

Line 190: by “day of dietary change” you mean D1H and D1G?

Line 192: please insert the number of the supplementary table.

Line 205: is this a sub-item of line 183?

Line 208: what does “p.” mean?

Figures S5 to S11 could be combined in one figure like you did in Fig 3. In fact, this new figure should be included as a main figure of the manuscript because its data (genus level) is much more meaningful than phylum level analysis.

Line 240: Table 4? I think this table along with OTUs data should be removed.

Line 242: different based on which test? LEfSe?

Line 256: please revise this statement. Alpha diversity indices do not take into account taxonomic information )membership), but just the number of different taxa.

Line 327: you are using only 6 animals in your study and this might be an overstatement.

Line 368: which diversity index was used to investigate diversity? It seems that only richness was included in your alpha diversity analysis.

Line 364–375: this paragraph is mainly repeating what was already said and might not be necessary.

One of the first studies using NGS to evaluate microbiota (maybe from Turnbaugh’s group) showed similar results: marked changes during the first day after abrupt diet change with subsequent recovery of the microbiota.

Ideally you should have a control group, bu if diet (or other environmental factors) was responsible for the observed microbiota changes, would you expect that the microbiota of D14G should be similar to D0? Was that the case?

6. PLOS authors have the option to publish the peer review history of their article (what does this mean?). If published, this will include your full peer review and any attached files.

Reviewer #1: No

---

## [Author Response · Author response to Decision Letter 0]

11 Jul 2020

A: I have added full affiliation including commune in France for one of the authors. I cannot find any other inconsistences with the template, could you please direct me to those if you notice anything else?

2. We noted in your submission details that a portion of your manuscript may have been presented or published elsewhere:

'An abstract based on the part of the current study was presented at EWEN (European Workshop in Equine Nutrition) 2018 in Sweden as oral presentation and published in the proceedings. This research study is a part of PhD thesis which is currently embargoed for 2 years. '

Please clarify whether this conference proceeding was peer-reviewed and formally published.

If this work was previously peer-reviewed and published, in the cover letter please provide the reason that this work does not constitute dual publication and should be included in the current manuscript.

The reason that this work does not constitute dual publication was included in the cover letter.

A: The following was included in the cover letter: “The abstract published in conference proceedings contains 350 words (0.5 page) compared to 7105 words (15 pages) in the current manuscript. To the best of our knowledge, it is a common practice to present preliminary results at the conferences before submitting complete manuscripts for publication. Based on our previous experience, this practice is usually well accepted by the journals. Please, also note, that the abstract published on conference proceedings does not repeat the abstract of this article.”

3. We note you have included tables to which you do not refer in the text of your manuscript. Please ensure that you refer to Tables 3 and 4 in your text; if accepted, production will need this reference to link the reader to the Table.

A: Reference to Table 3 and Table 4 was added (though table numbering was changed throughout the whole manuscript as reviewer has requested to replace one of the tables with figure).

Reviewers' comments:

Reviewer's Responses to Questions

Comments to the Author

1. Is the manuscript technically sound, and do the data support the conclusions?

Reviewer #1: Yes

2. Has the statistical analysis been performed appropriately and rigorously? 

Reviewer #1: Yes

3. Have the authors made all data underlying the findings in their manuscript fully available?

Reviewer #1: No

A: Please, check Datacite DOI: 10.5525/gla.researchdata.986 

If there anything else that needs to be included, could you please let me know what exactly that is?

4. Is the manuscript presented in an intelligible fashion and written in standard English?

Reviewer #1: No

 A: The manuscript has been checked for typographical and grammatical errors and corrections have been made. 

5. Review Comments to the Author

Reviewer #1: Dear Editor and authors.

This is a descriptive study evaluating changes in the fecal microbiota of 6 ponies undergoing diet change. The article is not completely novel, but brings interesting information to contribute to the current knowledge of factors affecting the intestinal microbiota of horses. The major limitation with the study design is the lack of control group that remained in pasture: variables such as temperature, pasture composition, change in environment (housing) could have affected their microbiota. This needs to be clearely addressed in the discussion. A: This was included and is now clearly addressed in the discussion. Also, authors give too much importance for the microbial variations at the phylum level, which is important, much less informative and meaningful. For example, which of the thousands species of Firmicutes increased after diet change? These genera should be presented in the main article (and not as supplementary data). As you will see in my comments below the article needs major rewriting to improve clarity.

Abstract:

The abstract needs major changes. You need to clearly state the study design including days of sampling, details on diet change, number of animals, etc. Many readers are not familiarized with concepts like LEfSe, LDA, PCoA, community structure, etc. You should use this space wisely to transmit the main findings and significance of your study in a way that most people will be able to understand it. A: The abstract was majorly re-written, unfamiliar to some of the readers concepts such as LEfSe, LDA, PCoA were excluded.

Line 53-56: I am not sure this statement should be used to justify your study. While fiber based diets likely benefit the horse microbiota, this is not feasible in performance horses in which high energy supplementation is required to fulfill their energy needs. A: This statement has been removed.

Line 59: please refer to Leclere and Costa, J Vet Intern Med. 2020 Mar;34(2):996-1006. doi: 10.1111/jvim.15748. A: Thanks, this was published March 2020, same month as we have submitted our manuscript, this is why I was not aware of this article on the day of submission. I have included this article to improve our discussion section and changed introduction to make it up to date.

Material and methods:

What was the composition and quality of hay (any analysis done?)? What was the grass type/variety? A: Unfortunately, nutritional analysis of the grass and hay were not performed. I have included this statement in Material and Methods to make it clear for the reader same as was done in a study of Leclere and Costa, J Vet Intern Med. 2020 Mar;34(2):996-1006. In fact, it is clear now to me that it would be better to do nutritional analysis of the feeds in all diet-related studies and not only digestibility studies. 

Table 1: it would be better to see that in a timeline rather than in a Table. You should not use the same nomenclature (e.g. D1) for both periods (e.g. line 170, 209, 294, etc.). In fact, every time you write “abrupt dietary change” you should specify to which of the changes you are referring to. A: Table 1 was replaced with Figure 1 (timeline). The nomenclature was checked and corrected throughout the whole manuscript.

Line 130: what is a phylotype? A: Operational Taxonomic Units (OTU) or phylotype, is usually used instead of “species” for a cluster of related 16S rRNA sequences in 16S rRNA sequencing studies. This was clarified in the text and reference provided.

Line 132: what is a OTU? Please define those terms. A: OTU was defined above, further explanation is given in discussion section. Did you use a phylotype or OTU approach? A: OTU approach

Line 147: adjusted using which test? A: Benjamini-Hochberg procedure, this was included in the manuscript

Can you comment on the reliability of those primers on classifying Archaea based on a scientific publication? It is my impression that primers designed to target bacteria are not adequate for Archaea analysis. A: This was expected because the primers were targeted on the V3/V4 region of bacterial rRNA subunit. V3/V4 region primers provide good general amplification, they are designed to pick up bacterial species not archaeal; however, they are not designed to specifically exclude archaea. 

Line 157: why were low abundance sequences excluded? Was this done before or after alpha diversity calculation? This will likely impact Chao index. A: Low abundance sequences were excluded to reduce the influence of rare genera/species on the overall results. Alpha rarefaction was performed before and after removal with very little difference. 

Results:

Line 160: is phylogenetic diversity a measure of Alpha diversity? A: PD_whole_tree is a phylogenetic metric of alpha-diversity. This is also know as Faith's Phylogenetic Diversity (PD_whole tree means phylogenetic diversity using the whole “tree”) (more is included in M&M section). Please indicate which test was used for that in the material and methods. A: Observed OTUs, Chao_1 and PD_whole_tree tests were used to assess alpha-diversity. This is included in M&M and corresponding Figures are presented in supplementary material (S1-S3) and Figure 1. 

Line 163: Please indicate the units after the values. A: Chao1 is based on the number of OTUs and is a measure of species richness. It has no units. The coherent unit for dimensionless quantities, also termed quantities of dimension one is symbol ”1” which is usually omitted. (Source: https://www.bipm.org/utils/common/pdf/si_brochure_8_en.pdf). These may also be referred to as arbitrary units.

Please provide details how you performed LEfSe: using the OTU table with all the taxonomic levels? (A: yes, this was including in the manuscript) using day of sampling as class (A: yes, day of sampling as class, this was including in the manuscript) (or samples from G and H analysed separately, or performed pairwise comparisons?). “LEfSe was performed using the OTU table with all the taxonomic levels and day of sampling as class.”

Wouldn’t make more sense to use only the different genera? A: This can be performed, but often misses higher level taxonomic features as biomarkers, for example where a family level marker is significantly differential but this is spread across several genera which are not independently differential. 

Results:

Line 154: counts of reads per sample or of OTUs per sample? A: Counts of reads per sample, this was clarified in the text.

Line 157: where are the rarefaction curves? A: The rarefaction curves are presented in Supplementary figures S1-S3. This analysis can be quite subjective and I recommend that Good’s coverage is presented instead. A: Good’s coverage is 1 - the number of singleton OTUs (single count) divided by the total abundance for all OTUs. Since all singletons are discarded during the analysis this metric is not really relevant or in fact calculable on this data.

Figure 1: could indicate the diets and day of change, just like you did in Figure 3. Please indicate what circles represent (outliers?). A: This was changed as requested.

Line 177: is the weighted Unifrac really evaluating microbial membership and structure or only structure. In my understanding, membership is evaluated by unweighted Unifrac. A: Unweighted uses the number of different species/genera to determine differences between samples. Weighted factors in abundances for each species/genera. In other words, in weighted UniFrac branch length were weighed by relative abundance, while unweighted UniFrac were based purely on sequence distance measure. So in one sense unweighted is a better estimator of membership if you are using a binary definition – i.e. it is or it isn’t). Unweighted UniFrac plots were added (Fig 3).

Line 180: OTUs A: Done

Figure 2: how can you distinguish the different days in figure 2A? Any clustering seen by individual?

A: Figure 2A (now 3A after numbering has changed) is there to show that there was no clustering by diet (grass vs hay) when all sampling days (G-D0, H-D1, H-D2, H-D3, H-D7, H-D14, G-D1, G-D2, H-D3, H-D7, H-D14) are taken into account. You cannot distinguish between sampling days, just between diets. Yes, the following pairs of individual animals were different: 1 and 6, 2 and 6, 3 and 5, 3 and 6, 5 and 6 (weighted UniFrac). Unweighted UniFrac: 1 and 3, 1 and 5, 2 and 3, 2 and 4, 2 and 5, 3 and 4, 3 and 5, 3 and 6, 4 and 5, 4 and 6, 5 and 6. I don’t think that adding this information would be relevant and add to the manuscript, but I can do that if you suggest it would be better to do so. 

Line 136: you mentioned that Illumina sequencing is not suitable for species level analysis (and you are completely right), so in my opinion it is not adequate to present this data, as it might mislead readers to erroneous information. 

A: I totally agree with what you say. Generally, we present genera level data. While it is not suitable for overall spp level analysis it can identify some spp accurately dependent on specific OTUs. Therefore, using the data to show species level differences is not particularly sound – but demonstrating the difference/presence of a one specific spp could be used. OTU data was previously used in other publications, but I removed it because I see your point. (Bulmer LS, Murray J-A, Burns NM, Garber A, Wemelsfelder F, McEwan NR, et al. High-starch diets alter equine faecal microbiota and increase behavioural reactivity. Scientific Reports. 2019;9(1):18621. 

Line 188: were those really unassigned organisms or it could be products of sequencing error not detected during bioinformatics? This should be acknowledged especially considering the approach adopted here (sequencing of the V3-V4 regions), which considerably decreases the overlapping and therefore, increases sequencing errors:

A: The level of sequencing error is not particularly high in Illumina data. Yes, some of these unassigned OTUs could be sequencing errors, they likewise could be novel bacteria, sequence variants or a number of other possibilities. The V3/V4 region is less than 450bp combined which means there is a 75bp overlap (even after trimming reads less than 250bp are discarded meaning the minimum overlap is 50bp) which is a very significant overlap therefore sequencing errors are unlikely.

https://mothur.org/blog/2014/Why-such-a-large-distance-matrix/

A: An interesting point and read, this is particularly important for their method of analysis (via Mothur) which clusters using the entire sequence. Qiime on the other hand clusters based on OTUs which minimises the influence of “off by one” and SNP variants eluded to in the blog post. 

Line 190: by “day of dietary change” you mean D1H and D1G? A: This was clarified, thanks. “Day of dietary change” was replaced with “sample collection days” (G-D0, H-D1, H-D2, H-D3, H-D7, H-D14, G-D1, G-D2, H-D3, H-D7, H-D14). Same nomenclature is now used in the whole manuscript and all tables and figures. 

Line 192: please insert the number of the supplementary table. A: This was done, thanks for noticing.

Line 205: is this a sub-item of line 183? A: Yes, thanks, this was corrected.

Line 208: what does “p.” mean? A: phylum, this was changed

Figures S5 to S11 could be combined in one figure like you did in Fig 3. In fact, this new figure should be included as a main figure of the manuscript because its data (genus level) is much more meaningful than phylum level analysis. A: This has been done now.

Line 240: Table 4? I think this table along with OTUs data should be removed. A: This has been removed.

Line 242: different based on which test? LEfSe? A: PERMANOVA, but this was deleted as reviewer has suggested

Line 256: please revise this statement. Alpha diversity indices do not take into account taxonomic information )membership), but just the number of different taxa. A: this statement has been revised and the following was included instead: “This indicated that the differences observed in the PCoA plots were likely due to differences in the RA of some taxonomic levels, as the numbers of taxa and the evenness does not change significantly between faecal samples from ponies fed either diet. This does not, however, exclude the possibility of a radical shift in taxa without an overall change in taxa numbers or evenness, although the taxonomic distribution strongly suggests this is not the case”.

Line 327: you are using only 6 animals in your study and this might be an overstatement. A: this amended according to your recommendation. “This hypothesis warrants further investigation using a larger animal sample size.” 

Line 368: which diversity index was used to investigate diversity? It seems that only richness was included in your alpha diversity analysis. A: PD_whole_tree was used as a measure of phylogenetic diversity. It was better explained in a M&M section.

Line 364–375: this paragraph is mainly repeating what was already said and might not be necessary. A part of this paragraph was deleted from the discussion section according to reviewer’s suggestion, another part was moved to the very end of the discussion section as it refers to future research that needs to be undertaken.

One of the first studies using NGS to evaluate microbiota (maybe from Turnbaugh’s group) showed similar results: marked changes during the first day after abrupt diet change with subsequent recovery of the microbiota. A: I think I found the article you were referring to and included it in our discussion section. Thanks a lot for your suggestion. (David LA, Maurice CF, Carmody RN, Gootenberg DB, Button JE, Wolfe BE, et al. Diet rapidly and reproducibly alters the human gut microbiome. Nature. 2014;505(7484):559-63. 

Ideally you should have a control group, bu if diet (or other environmental factors) was responsible for the observed microbiota changes, would you expect that the microbiota of D14G should be similar to D0? Was that the case? A: The comment regarding the absence of the control group remaining on pasture was added. If environmental factors were majorly responsible for the observed changes in microbial relative abundance, the microbiota of G-D14 most likely would not be similar to G-D0. In our study, according to LEfSe analysis there were no biomarkers identified for G-D0 or G-D14 (Table 2). According to weighted and unweighted UniFrac PCoA plots, there was no clustering observed on G-D0 and G-D14 (S6 Fig). Thus, it is likely that abrupt dietary changes were major factors leading to changes in faecal microbiota of the ponies abruptly transitioned from grass to hay and from hay to grass. (This was added to the manuscript).

---

## [Decision Letter · Decision Letter 1]

5 Aug 2020

Abrupt dietary changes between grass and hay alter faecal microbiota of ponies

PONE-D-20-08059R1

Dear Dr. Garber,

We’re pleased to inform you that your manuscript has been judged scientifically suitable for publication and will be formally accepted for publication once it meets all outstanding technical requirements.

Kind regards,

Juan J Loor

Academic Editor

PLOS ONE

Additional Editor Comments (optional):

Reviewers' comments:

Reviewer's Responses to Questions

**Comments to the Author**

1. If the authors have adequately addressed your comments raised in a previous round of review and you feel that this manuscript is now acceptable for publication, you may indicate that here to bypass the “Comments to the Author” section, enter your conflict of interest statement in the “Confidential to Editor” section, and submit your "Accept" recommendation.

Reviewer #1: All comments have been addressed

2. Is the manuscript technically sound, and do the data support the conclusions?

Reviewer #1: Yes

3. Has the statistical analysis been performed appropriately and rigorously? 

Reviewer #1: Yes

4. Have the authors made all data underlying the findings in their manuscript fully available?

Reviewer #1: Yes

5. Is the manuscript presented in an intelligible fashion and written in standard English?

Reviewer #1: Yes

6. Review Comments to the Author

Reviewer #1: Thank you very much for your clarifications and changes in the manuscript. It has largely improved the manuscript.

7. PLOS authors have the option to publish the peer review history of their article (what does this mean?). If published, this will include your full peer review and any attached files.

Reviewer #1: **Yes: **Marcio Costa

---

## [Editor Report · Acceptance letter]

10 Aug 2020

PONE-D-20-08059R1 

Abrupt dietary changes between grass and hay alter faecal microbiota of ponies 

Dear Dr. Garber:

I'm pleased to inform you that your manuscript has been deemed suitable for publication in PLOS ONE. Congratulations! Your manuscript is now with our production department. 

Kind regards, 

on behalf of

Dr. Juan J Loor 

Academic Editor

PLOS ONE